# Echo Planning for Autonomous Driving: From Current Observations to Future Trajectories and Back

## Abstract

Modern end-to-end autonomous driving systems suffer from a critical limitation: their planners lack mechanisms to enforce temporal consistency between predicted trajectories and evolving scene dynamics. This absence of self-supervision allows early prediction errors to compound catastrophically over time. We introduce Echo Planning (**EchoP**), a new self-correcting framework that establishes an end-to-end Current → Future → Current (CFC) cycle to harmonize trajectory prediction with scene coherence. Our key insight is that plausible future trajectories must be bi-directionally consistent, *i.e.*, not only generated from current observations but also capable of reconstructing them. The CFC mechanism first predicts future trajectories from the Bird's-Eye-View (BEV) scene representation, then inversely maps these trajectories back to estimate the current BEV state. By enforcing consistency between the original and reconstructed BEV representations through a cycle loss, the framework intrinsically penalizes physically implausible or misaligned trajectories. Experiments on nuScenes show that the proposed method yields competitive performance, reducing L2 error (Avg) by -0.04 m and collision rate by -0.12% compared to one-shot planners. The approach also scales to closed-loop evaluation, *i.e.*, Bench2Drive, attaining a 26.52% success rate. Notably, EchoP requires no additional supervision: the CFC cycle acts as an inductive bias that stabilizes long-horizon planning. Overall, EchoP offers a simple, deployable pathway to improve reliability in safety-critical autonomous driving.

## 1 Introduction

Vision-based end-to-end planning has become a leading paradigm in autonomous driving research (Hu et al., 2022; 2023; Jiang et al., 2023; Zheng et al., 2024; Sun et al., 2024; Weng et al., 2024; Li & Cui, 2025). Synchronized multi-view RGB images are usually mapped to Bird's-Eye-View (BEV) scene representation, providing a direct interface between perception and planning. For instance, dense BEV models rasterize the scene into grids and learn rich spatial features (Chen et al., 2024; Hu et al., 2022; 2023; Jiang et al., 2023; Zheng et al., 2024; Ye et al., 2023; Liu et al., 2023b; Zhang et al., 2024b), while sparse BEV models replace the grid with compact scene queries that reduce computation and label burden while preserving BEV geometry (Sun et al., 2024; Zhu et al., 2024; Li & Cui, 2025; Zhang et al., 2024a). Based on the BEV representation, most current approaches follow a one-shot planning paradigm, forecasting the vehicle's future path solely from a single snapshot of the present scene, shown in Fig 1 (a, b, c). However, these methods lack an internal mechanism to enforce temporal consistency. They generate trajectories without verifying if the predicted future states are consistent with the initial observation, causing early inaccuracies to compound over time and result in unsafe driving behaviors. Hence, an open research question persists: ***how can we inherently embed temporal consistency in trajectory planning without incurring additional annotation overhead or heuristic complexity?***

To address this challenge, we introduce **EchoP**, a novel self-correcting trajectory planning framework that enforces intrinsic temporal coherence through an end-to-end CFC feedback cycle (see Fig 1 (d)). While prior work (Li & Cui, 2025) models temporal variation by predicting a future BEV map from the current BEV and planned trajectory, they only enforce forward consistency and neglect

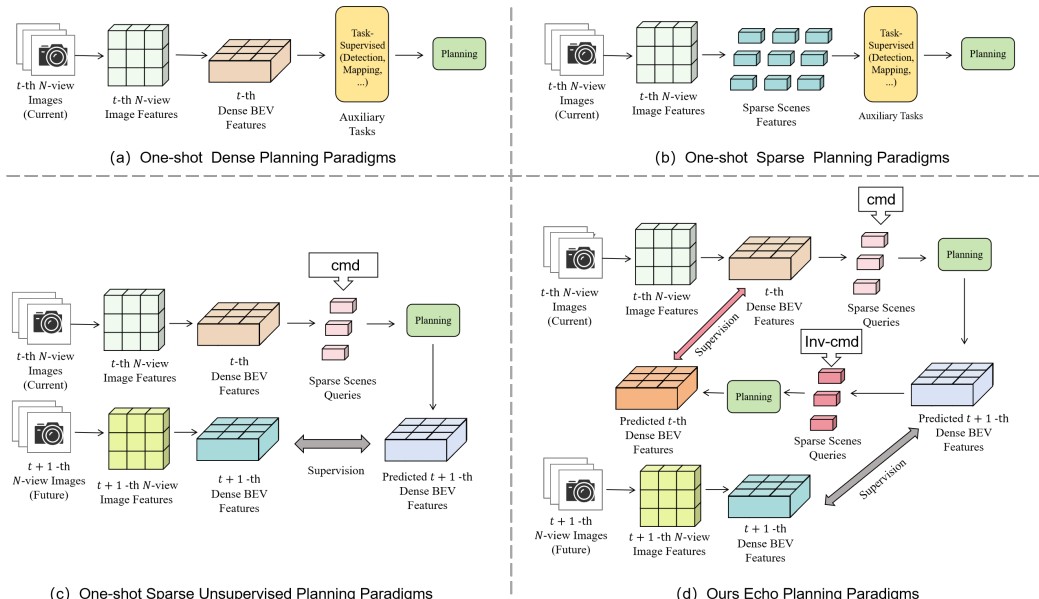

Figure 1: **Comparison between one-shot paradigms and our echo planning paradigms.** (a), (b), and (c) represent different types of one-shot approaches. Both (a) and (b) construct scene representations from image features and rely on auxiliary tasks to supervise the model, thereby enhancing environmental understanding; their main distinction lies in whether dense BEV features are used. Method (c) discards the auxiliary tasks introduced in (b) and is among the first to highlight the importance of temporal supervision, though it still considers only forward verification and thus remains within the one-shot paradigm. In contrast, ours (d) presents the echo planning approach, which employs a **Current → Future → Current** cycle. This design enforces **bidirectional self-supervision**, allowing the model to validate scene understanding without additional auxiliary tasks.

the reverse constraint. In contrast, the core intuition of our EchoP is that a robust trajectory must be bidirectionally consistent: it should not only represent a plausible future evolution of the current scene but also be able to accurately reconstruct the current scene when projected backward. Concretely, our model first generates future trajectories using a BEV representation of the current scene, and subsequently employs an inverse reconstruction step to predict back the current BEV state from these anticipated trajectories. This CFC cycle penalizes trajectories that are physically implausible or inconsistent with the initial scene configuration, thereby ensuring bidirectional consistency.

By enforcing the CFC feedback cycle, our method systematically reduces the impact of evolving scene dynamics on planning, preserves temporal consistency in the driving context, and consequently boosts trajectory accuracy. Experiments on the widely used nuScenes dataset (Caesar et al., 2020) confirm that our approach achieves competitive performance, sharply lowering both L2 trajectory error and collision rates relative to existing one-shot planners. Crucially, these gains are obtained without any additional supervision or handcrafted heuristics, making the method both reliable and readily deployable. Our EchoP therefore provides a systematic method to intrinsically enforce trajectory–scene consistency, advancing end-to-end autonomous driving toward safer and more dependable operation. Our primary contributions are:

- **Self-Correcting Planning.** We propose a novel echo planning paradigm for autonomous driving, built upon a CFC feedback loop. Unlike one-shot prediction methods, we leverage predicted future trajectories to inversely reconstruct the current Bird's-Eye-View (BEV) scene state. This reconstruction process enables implicit self-correction by penalizing implausible trajectories without requiring any external supervision.

- **High-Fidelity Driving Performance.** Extensive experiments on open-loop driving benchmarks, specifically nuScenes, validate that EchoP arrives at a competitive average collision rate of 0.17 without additional inference overhead. Moreover, EchoP can be seamlessly extended to closed-loop trajectory prediction, achieving a 26.52 success rate on the Bench2Drive benchmark.

## 2 RELATED WORK

**End-to-end planning.** Planning is the ultimate objective of the first phase of end-to-end autonomous driving. Early work relied on relatively simple neural networks that ignored much of the scene context and therefore offered limited interpretability (Pomerleau, 1988; Codevilla et al., 2018; Cheng et al., 2022; Bojarski et al., 2016; Codevilla et al., 2019; Prakash et al., 2021; Dauner et al., 2023). With the advent of large-scale datasets and stronger BEV perception, many learning-based planners have been proposed (Hu et al., 2022; Wu et al., 2022; Hu et al., 2023; Jiang et al., 2023; Zheng et al., 2024; Jia et al., 2024; 2023; Weng et al., 2024; Sun et al., 2024; Li et al., 2024b; Zhang et al., 2024a;b). Most recent systems still follow a perception, prediction, and planning pipeline to maintain transparency. ST-P3 (Hu et al., 2022) chains map perception, BEV occupancy prediction, and trajectory planning to infer future ego motion from surround cameras. UniAD (Hu et al., 2023) introduces a unified query design that integrates detection, mapping, and motion forecasting. VAD (Jiang et al., 2023) uses a vectorized scene representation to couple scene understanding with planning constraints, and GenAD (Zheng et al., 2024) generates future trajectories for both the ego vehicle and other agents within a learned probabilistic latent space. A newer line of research skips the explicit perception and prediction stages to facilitate efficiency. BEV-Planner (Li et al., 2024b) employs ego queries to extract task-relevant cues directly from BEV features, while SSR (Li & Cui, 2025) leverages navigation commands to focus on salient regions without perception supervision. However, little attention is paid to temporal consistency. We address this gap with an end-to-end CFC cycle that enforces coherent scene evolution over time.

**BEV perception in autonomous driving.** Perception forms the bedrock of autonomous driving, as we need distill actionable information from raw sensor streams. Precise and efficient scene understanding is therefore essential. The steady evolution of BEV representations (Chen et al., 2022; Hu et al., 2021a; Huang et al., 2021; Li et al., 2023; Liao et al., 2024a; 2022; Shao et al., 2023; Liu et al., 2023a; Zhang et al., 2022; Ju et al., 2025) has propelled perception forward, while the lower cost of RGB cameras is gradually supplanting LiDAR-based pipelines (Chen et al., 2023; Graham et al., 2018; Mao et al., 2021; Ye et al., 2023; Yuan et al., 2024). LSS (Philion & Fidler, 2020) is a landmark effort that used depth prediction to lift perspective features into BEV space. BEV-Former (Li et al., 2025) introduces spatial and temporal attention within a Transformer, achieving strong camera-only detection. To mitigate the high computational cost of dense BEV feature, sparse-perception methods have begun to emerge (Sun et al., 2024; Zhu et al., 2024; Zhang et al., 2024a; Li & Cui, 2025). SparseDrive (Sun et al., 2024) presents a symmetric sparse-perception module that jointly learns detection, tracking, and online mapping to produce a fully sparse scene representation. SparseAD (Zhang et al., 2024a) uses a compact query set to encode the driving scene sparsely. Our work follows this sparse-representation paradigm with zero annotation overhead.

## 3 METHOD

Given raw sensor inputs (*i.e.*, surrounding $N$-view camera images $I^i, i = 1, ..., N$) and high-level navigation command $C_{navi}$ extracted from the dataset, our Echo planning model plans the future trajectory $T$ of the ego Vehicle (see Fig 2). EchoP first encodes the image features with a backbone network and then uses bev queries to convert image features into BEV feature space (see Sec. 3.1). Second, EchoP distills task-critical cues from the ego-centric BEV space via a sparse scene representation module (Li & Cui, 2025) and condition them on the high-level navigation command, thereby mimicking the selective attention that human drivers pay to salient scene elements (Sec. 3.1). Finally, at the core of our approach, EchoP introduces a CFC cycle: it first predicts the ego-vehicle's future trajectory from the navigation-conditioned scene features, then inversely infers the present ego state from this forecast, and finally reconstructs the current BEV feature map (Sec. 3.2). Unlike prior end-to-end planners that propagate information only in the forward temporal direction, this bidirectional reasoning mechanism leverages temporal consistency within the ego sequence, leveraging the sequential context and yielding more reliable planning.

### 3.1 PRELIMINARY

**BEV features generation.** At the $t$-th time step, the surrounding $N$-view camera images $I^i, (i = 1, ..., N)$, are transformed into a BEV representation using BEVFormer (Li et al., 2025).

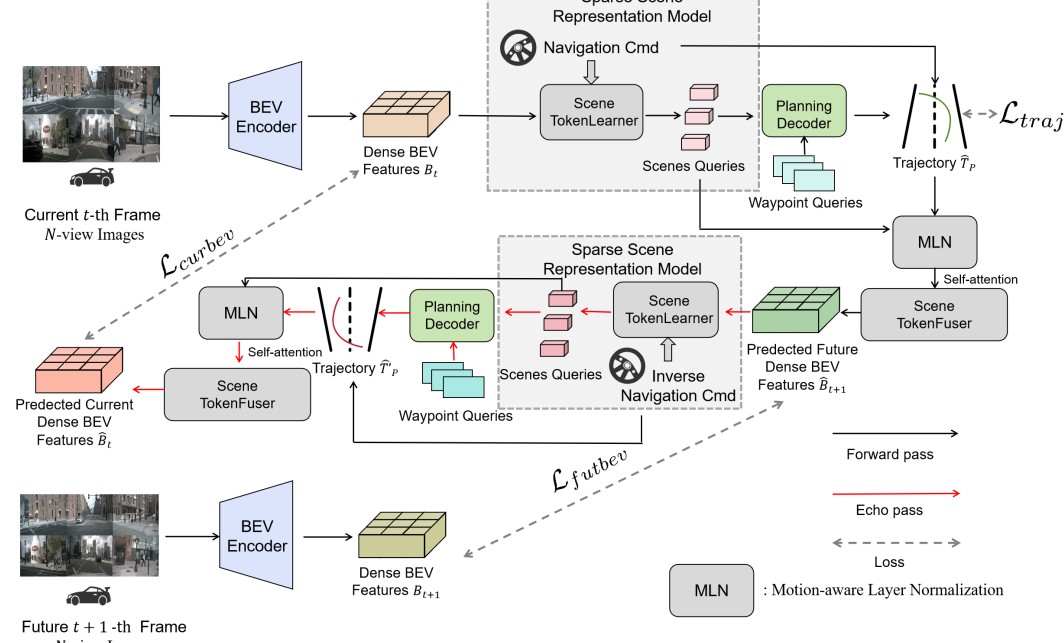

Figure 2: **The overview of our Echo Planning framework.** Echo Planning is trained through two complementary loops. The forward loop, indicated by black arrows in the figure, predicts future BEV features through the sparse scene representation model (Li & Cui, 2025) and applies self-supervision against the ground-truth BEV of future frames. The echo loop, shown in red, takes those predicted future features via the Motion-aware Layer Normalization (MLN) module (Wang et al., 2023; Li & Cui, 2025) and TokenFuser (Ryoo et al., 2021; Li & Cui, 2025), reconstructs the current BEV, and self-supervises against the ground-truth BEV of the current frame. By validating perception in both directions along the CFC cycle, the model cross-checks its understanding of the surrounding scene and thus produces reliable trajectory plans. **It is worth noting that the multiple modules, *e.g.*, Scene TokenLearner, Scene TokenFuser, Planning Decoder, and MLN, share weights during training. The CFC cycle does not impact inference speed, as only a forward pass from Current → Future is required during testing, consistent with existing methods.**

Specifically, sensor images $I^i$ first pass through a image backbone to yield image features $F_I^i, (i = 1, ..., N)$. Retaining the BEV representation $B_{t-1}$ from the preceding timestamp $t - 1$, a group of learnable BEV queries $Q \in \mathbb{R}^{H \times W \times K}$ is utilized to simultaneously extract temporal context from the previous BEV features $B_{t-1}$ and spatial information from the current multi-camera features $\left[F_I^i\right]_t$ via a cross attention mechanism. This results in the updated BEV representation $B_t \in \mathbb{R}^{H \times W \times K}$, where $H, W$ define the spatial resolution of the BEV plane, and $K$ denotes the feature channel dimensionality.

**Sparse scene representation module.** Given the BEV features and navigation command, the sparse scene representation module, such as SSR (Li & Cui, 2025), adopts the scene token learner to extract scene queries $S \in \mathbb{R}^{N_s \times K}$ from dense BEV features $B$, where $N_s$ is the number of scene queries. Firstly, the navigation command $C_{navi}$ is encoded into the dense BEV feature (Hu et al., 2018). Then, the navigation-aware BEV feature goes through the BEV TokenLearner module (Ryoo et al., 2021) and multi-layer self-attention (Vaswani et al., 2017) to get the current frame sparse scene representation $S_t$. After obtaining the current-frame scene representation $S_t$ that encodes the salient BEV context, we introduce a set of waypoint queries $W_t \in \mathbb{R}^{N_t \times N_c \times K}$. These queries attend to $S_t$ to capture the prospective motion of vehicle, where $N_t$ is the prediction horizon (in time steps), $N_c$ is the number of discrete navigation commands, and $K$ is the feature dimensionality. A lightweight multi-layer perceptron subsequently regresses the waypoint embeddings into planar coordinates, producing the predicted ego trajectory $T_P \in \mathbb{R}^{N_t \times 2}$ over the next $N_t$ time steps. For supervision, the high-level navigation command $C_{navi}$ selects the mode-consistent branch $\hat{T}_P$, which is compared with the ground-truth trajectory $T_{GT}$ using an L1 loss:

$$\mathcal{L}_{traj} = \left\| \hat{T}_P - T_{GT} \right\|_1. \tag{1}$$

### 3.2 ECHO PLANNING

**Motivation.** In end-to-end autonomous driving planning, real-time scene context must inform the prediction of the trajectory of the vehicle over the next few seconds, making temporal consistency indispensable. Beyond ensuring that the forecasted future state aligns with the true future trajectory, we also require that the current state, inferred in reverse from this prediction, matches the actual present state. Motivated by this bidirectional temporal constraint, we propose EchoP, a cycle-loop self-correcting framework that leverages self-supervised training across time to substantially enhance planning accuracy without requiring external supervision. As illustrated in Fig 2, we introduce a recurrent planner that forms a CFC BEV cycle loop, continuously validating trajectory predictions by inversely reconstructing the present BEV state.

**Forward pass (Current $\rightarrow$ Future).** The navigation-consistent trajectory prediction $\hat{T}_P$ and the current scene embedding $S_t$ are first processed by a Motion-aware Layer Normalization (MLN) module (Wang et al., 2023; Li & Cui, 2025), producing preliminary prediction queries. A self-attention block (Vaswani et al., 2017) then refines these queries into the future scene representation $\hat{S}_{t+1}$. Because the regions of interest in autonomous-driving scenes shift over time, we provide an additional source of supervision. The predicted future scene embedding $\hat{S}_{t+1}$ is fed to Token-Fuser (Ryoo et al., 2021; Li & Cui, 2025) to hallucinate a dense BEV feature map $\hat{B}_{t+1}$. We then enforce an L2 penalty between $\hat{B}_{t+1}$ and $B_t$ the ground-truth BEV, yielding the future-BEV reconstruction loss $\mathcal{L}_{futbev}$:

$$\mathcal{L}_{futbev} = \left\| \hat{B}_{t+1} - B_{t+1} \right\|_2 . \tag{2}$$

**Echo pass (Future $\rightarrow$ Current).** Our goal in this stage is to start from the predicted dense BEV at $T + 1$, $\hat{B}_{t+1}$, and run the pipeline in reverse to reconstruct the current BEV $\hat{B}_t$ for self-supervision. The Echo pass comprises four steps. Firstly, to mimic backward driving, we derive a reversed navigation command $C'_{navi}$ from the original $C_{navi}$ and feed $\left[ \hat{B}_{t+1}, C'_{navi} \right]$ into the same sparse-scene-representation module. The reversed navigation command $C'_{navi}$ is derived by inverting the original discrete command $C_{navi}$: specifically, the "left" and "right" directions are swapped, while "go straight" remains unchanged. Secondly, the scene token learner (Li & Cui, 2025), equipped with an attention module, extracts predicted future scene queries $\hat{S}'_{t+1}$ from the input BEV features $\hat{B}_{t+1}$. Thirdly, a new set of waypoint queries $W'_{t+1}$ attends to $\hat{S}'_{t+1}$, generating a reversed trajectory proposal $\hat{T}'_p$, after which the inverse command $C'_{navi}$ selects the mode-consistent branch of $\hat{T}'_p$. Finally, analogous to the forward pass, $\hat{T}'_p$ and $\hat{S}'_{t+1}$ are processed by an MLN block followed by stacked self-attention layers, producing the current-frame scene queries $\hat{S}'_t$. TokenFuser (Ryoo et al., 2021; Li & Cui, 2025) then reconstructs the dense BEV map $\hat{B}_t$ from $\hat{S}'_t$. The discrepancy between the reconstructed and ground-truth BEV features derives the current BEV reconstruction loss as:

$$\mathcal{L}_{curbev} = \left\| \hat{B}_t - B_t \right\|_2 . \tag{3}$$

**Total loss.** In summary, EchoP is trained end-to-end with a composite objective comprising the trajectory loss $\mathcal{L}_{traj}$, the future-BEV reconstruction loss $\mathcal{L}_{futbev}$, and the current-BEV reconstruction loss $\mathcal{L}_{curbev}$, combined as: $\mathcal{L}_{total} = \mathcal{L}_{traj} + \mathcal{L}_{futbev} + \mathcal{L}_{curbev}$.

## 4 EXPERIMENT

**Dataset.** We conduct extensive experiments on the widely adopted nuScenes dataset (Caesar et al., 2020) to assess the end-to-end planning capability of our EchoP framework. nuScenes comprises 1,000 driving logs, partitioned into 700, 150, and 150 scenes for training, validation, and testing, respectively. Each scene provides 20 seconds of synchronized RGB and LiDAR data captured at 12 Hz, together with 2 Hz key-frame annotations. Additionally, we compare our method with other methods on Bench2Drive (Jia et al., 2024), a closed-loop evaluation protocol under CARLA for end-to-end autonomous driving. We use the official 220 routes for evaluation.

**Evaluation metrics.** Following prior work, we evaluate with two metrics: L2 displacement error (the Euclidean distance between the planned and ground-truth trajectories) and collision rate (the

Table 1: Open-Loop Planning on nuScenes. The **top block** with $^o$ means the lidar-based methods. The **middle block** follows the UniAD protocol (final/max aggregation). The **lower block** with $^\ddagger$ follows VAD protocol (average over all predicted frames). $^\star$: The backbone is ResNet-101 He et al. (2016), while other methods without $^\star$ adopt ResNet-50 or similar variants. $^\S$: The result that we re-implement SSR with official weights Li & Cui (2025). The ↓ indicates that lower is better.

| Method | Auxiliary Task | L2 (m) ↓ | | | | Collision Rate (%) ↓ | | | |
|---|---|---|---|---|---|---|---|---|---|
| | | 1s | 2s | 3s | Avg. | 1s | 2s | 3s | Avg. |
| NMP$^o$ (Zeng et al., 2019) | Det & Motion | 0.53 | 1.25 | 2.67 | 1.48 | 0.04 | 0.12 | 0.87 | 0.34 |
| FF$^o$ (Hu et al., 2021b) | FreeSpace | 0.55 | 1.20 | 2.54 | 1.43 | 0.06 | 0.17 | 1.07 | 0.43 |
| EO$^o$ (Khurana et al., 2022) | FreeSpace | 0.67 | 1.36 | 2.78 | 1.60 | 0.04 | 0.09 | 0.88 | 0.33 |
| ST–P3 (Hu et al., 2022) | Det & Map & Depth | 1.72 | 3.26 | 4.86 | 3.28 | 0.44 | 1.08 | 3.01 | 1.51 |
| UniAD$^\star$ (Hu et al., 2023) | Det&Track&Map&Motion&Occ | 0.48 | 0.96 | 1.65 | 1.03 | 0.05 | 0.17 | 0.71 | 0.31 |
| OccNeXt$^\star$ (Tong et al., 2023) | Det & Map & Occ | 1.29 | 2.13 | 2.99 | 2.14 | 0.21 | 0.59 | 1.37 | 0.72 |
| VAD–Base (Jiang et al., 2023) | Det & Map & Motion | 0.54 | 1.15 | 1.98 | 1.22 | 0.04 | 0.39 | 1.17 | 0.53 |
| PARA–Drive (Weng et al., 2024) | Det&Track&Map&Motion&Occ | 0.40 | 0.77 | 1.31 | 0.83 | 0.07 | 0.25 | 0.60 | 0.30 |
| GenAD (Zheng et al., 2024) | Det & Map & Motion | 0.36 | 0.83 | 1.55 | 0.91 | 0.06 | 0.23 | 1.00 | 0.43 |
| UAD–Tiny (Guo et al., 2024) | Det | 0.47 | 0.99 | 1.71 | 1.06 | 0.08 | 0.39 | 0.90 | 0.46 |
| UAD$^\star$ (Guo et al., 2024) | Det | 0.39 | 0.81 | 1.50 | 0.90 | **0.01** | 0.12 | 0.43 | 0.19 |
| SSR$^\S$ (Li & Cui, 2025) | None | 0.25 | 0.64 | 1.33 | 0.74 | 0.16 | 0.21 | 0.51 | 0.29 |
| **EchoP (Ours)** | None | **0.23** | **0.60** | **1.27** | **0.70** (-0.04) | 0.02 | **0.12** | **0.39** | **0.17** (-0.12) |
| ST–P3$^\ddagger$ (Hu et al., 2022) | Det & Map & Depth | 1.33 | 2.11 | 2.90 | 2.11 | 0.23 | 0.62 | 1.27 | 0.71 |
| UniAD$^{\star\ddagger}$ (Hu et al., 2023) | Det&Track&Map&Motion&Occ | 0.44 | 0.67 | 0.96 | 0.69 | 0.04 | 0.08 | 0.23 | 0.12 |
| VAD–Tiny$^\ddagger$ (Jiang et al., 2023) | Det & Map & Motion | 0.46 | 0.76 | 1.12 | 0.78 | 0.21 | 0.35 | 0.58 | 0.38 |
| VAD–Base$^\ddagger$ (Jiang et al., 2023) | Det & Map & Motion | 0.41 | 0.70 | 1.05 | 0.72 | 0.07 | 0.17 | 0.41 | 0.22 |
| HE–Drive$^\ddagger$ (Wang et al., 2024) | Det&Track&Map | 0.31 | 0.58 | 0.93 | 0.60 | 0.01 | 0.05 | 0.16 | 0.07 |
| BEV–Planner$^\ddagger$ (Li et al., 2024b) | None | 0.28 | 0.42 | 0.68 | 0.46 | 0.04 | 0.37 | 1.07 | 0.49 |
| PARA–Drive$^\ddagger$ (Weng et al., 2024) | Det&Track&Map&Motion&Occ | 0.25 | 0.46 | 0.74 | 0.48 | 0.14 | 0.23 | 0.39 | 0.25 |
| LAW$^\ddagger$ (Li et al., 2024a) | None | 0.26 | 0.57 | 1.01 | 0.61 | 0.14 | 0.21 | 0.54 | 0.30 |
| GenAD$^\ddagger$ (Zheng et al., 2024) | Det & Map & Motion | 0.28 | 0.49 | 0.78 | 0.52 | 0.08 | 0.14 | 0.34 | 0.19 |
| SparseDrive$^\ddagger$ (Sun et al., 2024) | Det&Track&Map&Motion | 0.29 | 0.58 | 0.96 | 0.61 | 0.01 | 0.05 | 0.18 | 0.08 |
| UAD$^{\star\ddagger}$ (Guo et al., 2024) | Det | 0.28 | 0.41 | 0.65 | 0.45 | **0.01** | **0.03** | 0.14 | **0.06** |
| DiffDrive$^\ddagger$ (Liao et al., 2024b) | Det &Map | 0.27 | 0.54 | 0.90 | 0.57 | 0.03 | 0.05 | 0.16 | 0.08 |
| SSR$^{\S\ddagger}$ (Li & Cui, 2025) | None | 0.19 | 0.36 | 0.62 | 0.39 | 0.10 | 0.13 | 0.22 | 0.15 |
| **EchoP$^\ddagger$ (Ours)** | None | **0.17** | **0.33** | **0.58** | **0.36** (-0.03) | 0.03 | 0.05 | **0.14** | 0.07 (-0.08) |

Table 2: Open-loop and Closed-Loop Planning on Bench2Drive. The ↑ indicates that lower is better. The ↓ indicates that lower is better. $^\S$: The result that we retrain the SSR (Li & Cui, 2025).

| | Open-loop | Closed-loop | | |
|---|---|---|---|---|
| Method | Avg. L2 ↓ | Driving score ↑ | Success Rate (%) ↑ | Efficiency ↑ |
| AD-MLP (Zhai et al., 2023) | 3.64 | 18.05 | 0.00 | 48.45 |
| UniAD-Tiny (Hu et al., 2023) | 0.80 | 40.73 | 13.18 | 123.92 |
| UniAD-Base (Hu et al., 2023) | **0.73** | 45.81 | 16.36 | 129.21 |
| VAD (Jiang et al., 2023) | 0.91 | 42.35 | 15.00 | 157.94 |
| GenAD (Zheng et al., 2024) | - | 44.81 | 15.90 | - |
| SSR$^\S$ (Li & Cui, 2025) | 0.90 | 32.34 | 11.85 | 157.62 |
| EchoP (ours) | 0.79 | **50.35** | **26.54** | **185.18** |

proportion of planned trajectories that collide with any traffic participant). Unless specified otherwise, the model is fed with 2 s of history corresponding to five frames, and planning quality is reported for 1 s, 2 s, and 3 s horizons. Because VAD (Jiang et al., 2023) and UniAD (Hu et al., 2023) adopt different aggregation rules, we report both results. VAD averages over all predicted frames, whereas UniAD uses the final value (maximum collision). Table 1 presents the outcomes under both protocols.

**Implementation details.** EchoP is trained for 12 epochs on 4 RTX-A6000 GPUs with a batch size of 1 per GPU. The image encoder is a ResNet-50 (He et al., 2016) fed with $640 \times 360$ images,

Table 3: Ablation for our CFC cycle. The upper block follows the UniAD protocol (final/max aggregation). The lower block with ‡ follows VAD protocol (average over all predicted frames).§: The result that we retrain the SSR (Li & Cui, 2025) on 4 GPUs. The ↓ indicates that lower is better.

| Method | CFC cycle | L2 (m) ↓ | | | | Collision Rate (%) ↓ | | | |
|---|---|---|---|---|---|---|---|---|---|
| | | 1s | 2s | 3s | Avg. | 1s | 2s | 3s | Avg. |
| SSR§ (Li & Cui, 2025) | | 0.25 | 0.64 | 1.33 | 0.74 | 0.16 | 0.21 | 0.51 | 0.29 |
| **EchoP (Ours)** | ✓ | **0.23** | **0.60** | **1.27** | **0.70** | **0.02** | **0.12** | **0.39** | **0.17** |
| SSR§‡ (Li & Cui, 2025) | | 0.19 | 0.36 | 0.62 | 0.39 | 0.10 | 0.13 | 0.22 | 0.15 |
| **EchoP‡ (Ours)** | ✓ | **0.17** | **0.33** | **0.58** | **0.36** | **0.03** | **0.05** | **0.14** | **0.07** |

Table 4: Ablation for the CFC cycle effectiveness in different scene representation numbers. The block all follows the UniAD protocol (final/max aggregation). The row with a gray background is the re-implementation of our baseline. The ↓ indicates that lower is better.

| $N_s$ | CFC cycle | L2 (m) ↓ | | | | Collision Rate (%) ↓ | | | |
|---|---|---|---|---|---|---|---|---|---|
| | | 1s | 2s | 3s | Avg. | 1s | 2s | 3s | Avg. |
| 8 | | 0.27 | 0.69 | 1.43 | 0.79 | 0.20 | 0.35 | 0.68 | 0.41 |
| 8 | ✓ | **0.21** | **0.57** | **1.22** | **0.67** | **0.00** | **0.06** | **0.45** | **0.17** |
| 16 | | 0.25 | 0.64 | 1.33 | 0.74 | 0.16 | 0.21 | 0.51 | 0.29 |
| 16 | ✓ | **0.23** | **0.60** | **1.27** | **0.70** | **0.02** | **0.12** | **0.39** | **0.17** |

Table 5: Temporal consistency on the test set. We measure the $\mathcal{L}_{curbev} = \left\| \hat{B}_t - B_t \right\|_2$. Lower is better. We could observe that our method ensure the bidirectional consistency on the unseen test set.

| Method | $\mathcal{L}_{curbev}$ ↓ |
|---|---|
| Baseline | 0.2442 |
| EchoP (CFC cycle) | **0.0962** |

Table 6: Ablation for the loss weight. Varying $\mathcal{L}_{curbev}$ and $\mathcal{L}_{futbev}$.

| $\mathcal{L}_{curbev}$ | $\mathcal{L}_{futbev}$ | L2 (m) ↓ | | | | Collision Rate (%) ↓ | | | |
|---|---|---|---|---|---|---|---|---|---|
| | | 1s | 2s | 3s | Avg. | 1s | 2s | 3s | Avg. |
| 0.1 | 0.5 | **0.23** | **0.60** | **1.27** | **0.70** | 0.02 | 0.12 | 0.39 | 0.17 |
| 0.5 | 0.5 | 0.24 | 0.63 | 1.34 | 0.73 | **0.00** | **0.10** | **0.35** | **0.15** |
| 0.1 | 0.8 | 0.24 | 0.63 | 1.31 | 0.72 | 0.04 | 0.14 | 0.45 | 0.21 |
| 0.1 | 0.5 | **0.17** | **0.33** | **0.58** | **0.36** | **0.03** | **0.05** | **0.14** | **0.07** |
| 0.5 | 0.5 | 0.19 | 0.36 | 0.64 | 0.39 | 0.08 | 0.06 | 0.13 | 0.09 |
| 0.1 | 0.8 | 0.18 | 0.35 | 0.61 | 0.38 | 0.00 | 0.05 | 0.21 | 0.09 |

yielding a 100 × 100 BEV grid and 16 × 256 sparse tokens. The navigation-command set remains 3, matching prior work. For training, optimization uses AdamW (Loshchilov & Hutter, 2017) with a learning rate of $5 \times 10^{-5}$, and all other hyperparameters follow SSR (Li & Cui, 2025). Unless stated, loss weights for trajectory, future-BEV, and current-BEV are 1.0, 0.5, and 0.1 by default.

### 4.1 COMPARISON WITH THE STATE-OF-THE-ART METHODS

**Open-Loop Evaluation.** To assess the effectiveness of Echo Planning, we train and evaluate the model on the widely used nuScenes benchmark. As summarised in Table 1, Echo Planning achieves the best results among contemporary end-to-end planners on both the L2 error and the collision rate, attaining an average $L2_{MAX}$ error of only 0.70 m. Compared to strong one-shot planners, our approach delivers consistent benefits. Against UniAD the average $L2_{MAX}$ error drops by 0.33 m (32% relatively) and the average $CR_{MAX}$ falls by 0.14% (45% relatively). Compared with VAD-Base, the gains are even larger, with reductions of 0.52 m in average $L2_{MAX}$ (43% relatively) and 0.36% in average $CR_{MAX}$ (68% relatively). In addition, we compare our approach to the SSR baseline. Due to recent updates in the SSR codebase, our reproduced results on a 4-GPU setup differ from those reported in the original paper, a point also acknowledged by the authors on GitHub, as their modifications affected collision rate performance. For a fair comparison, we benchmark Echo Planning against our reproduced SSR results and observe substantial improvements in both L2 error and collision rate. Echo Planning still lowers the average $L2_{MAX}$ by 0.04 m and the average $CR_{MAX}$ by 0.12%, while also improving the $L2_{AVG}$ and $CR_{AVG}$ metrics by 0.03 m and 0.08% respectively. The consistent improvements across all indicators confirm that the Current → Future → Current cycle provides a reliable mechanism for accurate and safe trajectory planning.

**Closed-Loop Evaluation.** To evaluate the effectiveness of our approach under closed-loop control, we follow the Bench2Drive benchmark Jia et al. (2024). As shown in Table 2, EchoP markedly surpasses the SSR baseline (Li & Cui, 2025) in Driving Score. Relative to widely used methods, EchoP attains +4.54 and +8.00 points over UniAD (Hu et al., 2023) and VAD (Jiang et al., 2023), respectively. With the proposed CFC cycle enabled, EchoP further lifts the closed-loop Success Rate by +10.18% compared to UniAD (Hu et al., 2023). Moreover, because EchoP does not rely on auxiliary tasks, it achieves superior runtime efficiency while maintaining high success rates.

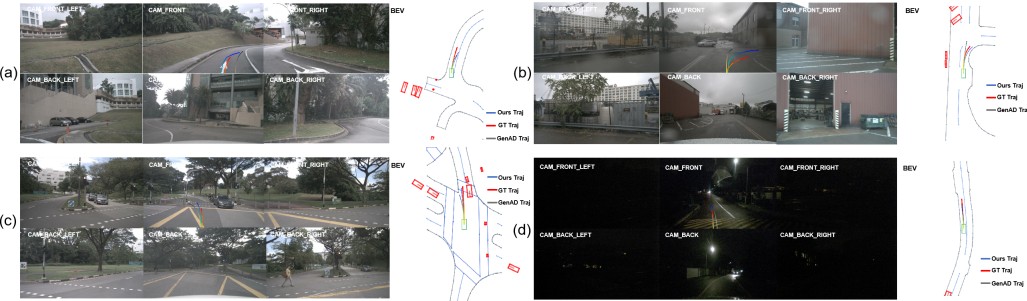

Figure 3: **Visualization of planning trajectories in different scenes.** We visualize several driving scenes, overlaying the trajectories predicted by our method alongside those of one of the one-shot planners, *i.e.*, GenAD (Zheng et al., 2024), and the ground-truth trajectories. Map context is rendered directly from the dataset annotations. In each figure, the green icon denotes the ego vehicle, while red icons mark surrounding vehicles and other dynamic objects. Video is in **Suppl.**

## 4.2 ABLATION STUDIES AND FURTHER DISCUSSION

**Effect of the CFC cycle.** In Table 3 we present an ablation of the CFC BEV cycle loop. Applying the CFC cycle to our one-shot planners reduces the average $L2_{MAX}$ error by 0.04 m and lowers the average $CR_{MAX}$ by 0.12%. At the same time, the average $L2_{AVG}$ error falls by 0.03 m and the average $CR_{AVG}$ drops by 0.08%. These results highlight the significant benefit of introducing bidirectional validation through inverse reconstruction, thereby improving model understanding of dynamic driving scenes. Our Echo Planning framework consequently enables safer trajectory generation and leads to substantial improvements in planning. To quantitatively assess the impact of our proposed CFC cycle on temporal consistency, we compare the Mean Squared Error (MSE) between the Current → Future→ Current predicted BEV and the Current BEV on the test set. As shown in Table 5, we conduct this comparison using models trained with the baseline method and our EchoP method (with CFC cycle).

**Effect of different scene representation numbers.** In Table 4, we assess the robustness of the CFC cycle when the scene representation is compressed to different capacities. We compare two settings: 8 and 16 scene tokens, while keeping all other factors fixed. In both cases, the CFC-augmented model secures clear improvements over its non-cycle counterpart. Remarkably, even with only 8 tokens, our method can still learn the surrounding environment of the vehicle very well, confirming that the cycle constraint remains effective even in highly compact representations.

**Effect of the loss weight.** As shown in Table 6, we show an ablation study on the weights of our three loss terms, with emphasis on the two components tied to the CFC BEV cycle. Keeping the trajectory loss $\mathcal{L}_{traj}$ fixed, we systematically vary the forward-pass loss $\mathcal{L}_{futbev}$ and the echo-pass loss $\mathcal{L}_{curbev}$. The results show that planning quality remains consistently high across a broad range of weight settings, underscoring the stability imparted by the CFC cycle and confirming that EchoP reliably strengthens planning performance.

**Why the end-to-end CFC BEV cycle loop effective?** Accurate trajectory planning requires the model to form a faithful understanding of the driving scene. Most recent planners follow a one-shot paradigm: the network observes a single sensor frame, predicts the future trajectory, and is trained either with auxiliary labels such as detection or mapping or with a forward self-supervised loss that compares the predicted future scene to ground truth. This design leaves two critical gaps. First, heavy auxiliary heads greatly increase model complexity while offering no direct feedback on how well the planner actually understands its surroundings. Second, relying only on forward supervision ignores the reverse constraint: because the driving scene evolves unpredictably, a plan generated from the present frame alone may remain unverified and can drift from reality. Echo Planning addresses these issues with a CFC cycle. Without any extra supervision, the framework not only learns from the forward prediction of the future scene but also reconstructs the current BEV from that prediction, thereby checking whether the imagined future is consistent with the observed present. This bidirectional loop provides an intrinsic error detector that continuously validates trajectories and significantly lowers the probability of collisions, yielding more reliable planning.

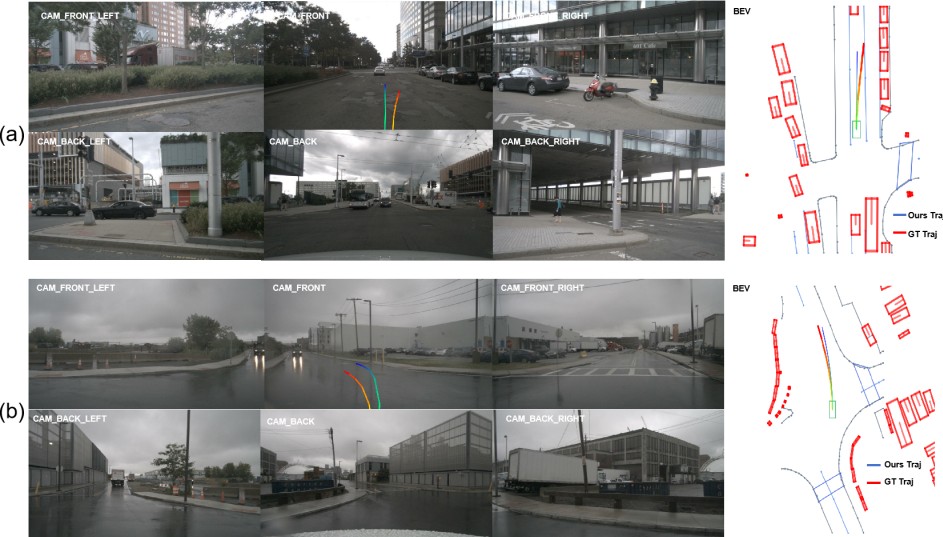

Figure 4: **Visualization of failure cases.** We highlight two distinct planning failure cases. The first arises when the navigation cue in the ground-truth annotation is ambiguous, and the second occurs in open spaces where the predicted trajectory deviates during a turn. In each figure, the green icon denotes the ego vehicle, while red icons mark surrounding vehicles and other dynamic objects.

### 4.3 QUALITATIVE RESULTS

**Visualization of planning trajectories.** In Figure 3, we qualitatively compare trajectories planned by EchoP with ground truth (GT) and GenAD (Zheng et al., 2024) across four representative scenarios. Figure 3 (a) depicts an intersection requiring a left turn. EchoP generates a smoother and more anticipatory turning arc than the GT trajectory, aligning better with the lane geometry. Figure 3 (b) considers driving conditions under rain-induced poor visibility. Despite water puddles and reflections obscuring lane markings, EchoP accurately infers the roadway layout, generating a right-turn trajectory more faithful to the lane than the GT. Figure 3 (c) illustrates navigation through a complex multi-way junction. Here, EchoP closely follows the GT trajectory without crossing road boundaries, whereas GenAD drifts into non-drivable areas. Figure 3 (d) shows a scenario of low-light night driving. Under dim illumination, EchoP closely matches the GT trajectory, while GenAD exhibits significant deviations. These examples validate that the CFC cycle in EchoP intrinsically corrects trajectory predictions, enhancing robustness across diverse conditions.

**Failure cases.** In Figure 4, we show two representative failure modes. The first stems from ambiguous navigation cues in the nuScenes annotation: in Fig 4 (a), the annotation marks the command as "go straight," yet the ground-truth trajectory actually includes a rightward lane change. When such inconsistencies arise, the planner can be misled and produce a biased path. The second limitation appears on very wide roads that lack explicit lane markings. Although the CFC cycle improves scene understanding, an open roadway grants considerable freedom even for human drivers. As a result, EchoP generates a trajectory that deviates from the ground truth, as in Fig 4 (b), while still remaining within a plausible driving corridor and preserving the correct turning direction.

## 5 CONCLUSION

The Echo Planning framework introduces a self-correcting Current → Future → Current BEV cycle loop (CFC cycle) that endows end-to-end planners with an intrinsic notion of temporal consistency. By predicting a future trajectory, inverting it to reconstruct the present BEV, and penalising any mismatch through a cycle loss, the framework detects and suppresses implausible motion without extra labels or auxiliary heads. Extensive nuScenes and Bench2Drive experiments confirm that this bidirectional feedback reduces displacement error and collision rate beyond state-of-the-art one-shot planners, while incurring no annotation overhead and minimal computation. Echo Planning, therefore, offers a practical, safety-oriented upgrade for modern autonomous-driving stacks, and we expect its cycle-consistency principle to extend naturally to longer horizons, richer modalities, and more closed-loop simulation in future work.

ETHICS STATEMENT

We hereby solemnly declare that **we have carefully read the ICLR Code of Ethics, and that this research strictly adheres to these guidelines**.

REPRODUCIBILITY STATEMENT

We are committed to ensuring the reproducibility of this work. To ensure the reproducibility of our research, we will provide comprehensive code, data, and experimental details.

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
