# OpenReview forum: "Echo Planning for Autonomous Driving: From Current Observations to Future Trajectories and Back"
_ICLR.cc/2026/Conference — ICLR 2026 Conference Withdrawn Submission_

### Official Review · Reviewer_zGu8 · 2025-10-22

**Soundness:** 3
**Presentation:** 3
**Contribution:** 2
**Rating:** 2
**Confidence:** 5

**Summary:**

This paper proposes an end-to-end planning algorithm that predicts future trajectories to generate corresponding future BEV features. It then reconstructs the current BEV features from these predictions and applies a consistency constraint between the reconstructed BEV and the BEV features encoded from multi-view images, leading to improved planning performance.

**Strengths:**

1. The proposed strategy EchoP is clearly described and well visualized, making the algorithm easy to understand.
2. The paper conducts quantitative experiments on the nuScenes and Bench2Drive datasets, showing noticeable improvements on Bench2Drive.

**Weaknesses:**

1. In the abstract, Figure 1(d), and Table 1, the paper seems to give the impression that the proposed method is self-supervised and does not rely on auxiliary task supervision (such as detection or mapping) shown in Figure 1(a,b). However, as I understand it, the BEV module still depends on these supervised tasks. This is also mentioned in line 161: “…are transformed into a BEV representation using BEVFormer…”, where BEVFormer indeed relies on such supervision to generate BEV features. Therefore, I think the related statement about "self-supervision" could be somewhat misleading.
2. The proposed approach involves forward and backward BEV feature prediction (resulting in four BEV feature maps in total). Since these predictions inevitably contain errors, it is questionable whether performing supervision on these imperfect BEV features can truly provide a benefit. Although ablation studies are provided, the results on the nuScenes dataset (which is dominated by straight-driving scenarios) are not very convincing.
3. When reconstructing the current BEV features based on predicted future BEV features, ego-motion naturally leads to blank or noisy regions near the boundaries (areas unobservable at the current time). Using these imperfect features to reconstruct the current BEV may cause information loss or distortion. Consequently, aligning these with the BEV features encoded directly from surround-view images may be unreliable, which raises doubts about the effectiveness of such a supervision strategy.

**Questions:**

1. I find the motivation of this paper difficult to understand. It is unclear why predicting future BEV features based on the predicted trajectory, then reconstructing the current BEV features from these predicted future features (conditioned on the inverted predicted trajectory), and finally enforcing consistency with the real current BEV features would lead to improved performance. The underlying reasoning for why this bidirectional prediction and consistency supervision should provide meaningful benefits is not well justified.
2. In line 224, the paper states that the reverse instruction “go straight” remains unchanged. However, this could be ambiguous. During the backward reconstruction of BEV features, the instruction actually refers to moving straight backward, not forward. If the same instruction token is used for both forward and backward processes, it is unclear how the SceneTokenLearner distinguishes between these two different contexts.

---

### Official Review · Reviewer_Qv8y · 2025-10-24

**Soundness:** 2
**Presentation:** 2
**Contribution:** 2
**Rating:** 4
**Confidence:** 5

**Summary:**

This paper introduces EchoP, a self-correcting trajectory planning framework for autonomous driving that enforces bidirectional temporal consistency through a Current→Future→Current (CFC) feedback cycle; unlike conventional one-shot planners that predict a future trajectory from a single scene snapshot, EchoP first generates a future trajectory from the current Bird's-Eye-View (BEV) representation and then inversely reconstructs the current BEV state from this predicted future, using the discrepancy between the reconstructed and actual current BEV as a self-supervised signal to penalize implausible trajectories. This CFC cycle, which requires no external supervision or auxiliary tasks, significantly enhances planning accuracy and safety, as demonstrated by state-of-the-art results on the nuScenes and Bench2Drive benchmarks.

**Strengths:**

1. Overall presentation: The paper is well-organized and clearly presented, with informative figures, well-structured tables, and clear writing. The overall presentation quality makes the paper easy to follow and understand.

2. Experiments and visualization: The experimental section is comprehensive. The visualizations are clear and intuitive, effectively demonstrating the qualitative performance.

**Weaknesses:**

1. **Limited novelty**: This paper is highly similar to SSR [1] and LAW [2], with only minor modifications to the loss calculation for the predicted current BEV feature. The overall contribution appears largely story-driven and incremental, which may not meet the novelty threshold for a top-tier conference such as ICLR.

2. **Unconvincing experiments**: The experiments on the nuScenes dataset have been widely questioned by previous works [3], and the results on the Bench2Drive benchmark are inferior to several recent methods, such as DriveTransformer [4]. Therefore, the experimental evidence does not sufficiently support the paper’s claims.

**References**:

[1] Navigation-Guided Sparse Scene Representation for End-to-End Autonomous Driving, ICLR 2025.

[2] Enhancing End-to-End Autonomous Driving with Latent World Model, ICLR 2025.

[3] Is Ego Status All You Need for Open-Loop End-to-End Autonomous Driving?, CVPR 2024.

[4] DriveTransformer: Unified Transformer for Scalable End-to-End Autonomous Driving, ICLR 2025.

**Questions:**

See weaknesses.

---

### Official Review · Reviewer_AwKP · 2025-10-25

**Soundness:** 2
**Presentation:** 2
**Contribution:** 2
**Rating:** 2
**Confidence:** 4

**Summary:**

This paper introduces Echo Planning (EchoP), a self-correcting framework for end-to-end autonomous driving that enforces temporal consistency via a Current → Future → Current (CFC) cycle. The key idea is that predicted trajectories should not only be consistent with future states but also be able to reconstruct the current scene, thereby acting as an implicit consistency check. EchoP achieves this by predicting future trajectories from BEV features, then reconstructing the current BEV from those predictions with inverse driving command, and penalizing inconsistencies. The method is evaluated on nuScenes and Bench2Drive, showing performance improvements without requiring additional supervision.

**Strengths:**

- Sufficient technical details. The CFC cycle is a creative and intuitive way to enforce bidirectional temporal consistency without extra labels, distinguishing it from prior one-shot planners.

- Strong empirical results. EchoP achieves SOTA performance on both open-loop (nuScenes) and closed-loop (Bench2Drive) benchmarks, with notable collision rate reductions.

- Exhaustive Ablation studies. Comprehensive ablations on loss weights, token numbers, and CFC cycle inclusion demonstrate methodological rigor and effectiveness of module design.

- Qualitative insights: Provided visualizations effectively illustrate how EchoP corrects trajectory drift and handles challenging driving scenes (e.g., low-light, intersections).

**Weaknesses:**

- Limited novelty. The CFC cycle is conceptually interesting, but the core modules (BEVFormer, TokenLearner, TokenFuser, MLN) are borrowed from prior works. It seem that the main contribution of EchoP is the additional branch for reversed reconstruction compared to SSR, which is somewhat limited.

- Underexplored theoretical grounding: The bidirectional consistency idea is intuitive but lacks formal justification or comparison with alternative consistency losses (e.g., contrastive, adversarial).

- Missing comparisons with new baselines. Some baselines (e.g., NMP, FF, EO) are outdated. The gain over SSR, the closest baseline, is marginal (e.g., 0.04m L2 improvement). Recent works such as DriveTransformer and GoalFLow with advanced performance should be included for further comparisons and discussions.

- Incomplete ablations. No ablation on reverse navigation command design (e.g., why simply flipping left/right is sufficient). No comparison with alternative cycle designs (e.g., Future → Current → Future). No analysis of failure modes beyond ambiguous nav cues or open roads. Besides, most of ablations are conducted on nuScenes dataset, where the open-loop metrics maybe somewhat unconvinced under closed-loop settings, and the performance gains among different designs are incremental, hindering the credibility of the proposed method.

- Writing and presentation issues. Inconsistent use of “EchoP” vs. “Echo Planning”.

**Questions:**

1. What makes the CFC cycle superior to forward-only or future-conditioned losses? Has alternatives like future-to-current reconstruction been explored?

2. Is the reverse navigation command (flipping left/right) sufficient for modeling backward dynamics? How does this handle U-turns or complex maneuvers?

3. Why does LiDAR-free, camera-only BEV suffice for bidirectional reconstruction? Is the geometry preservation under inversion guaranteed? Is there any error evaluations on the reconstructed performance of BEV features?

4. The cycle loss is applied in feature space (BEV) explicitly, rather than trajectory space. How does this ensure trajectory plausibility?

5. Why is batch size = 1 per GPU used, which maybe harmful for stable training. Is this due to memory constraints or design choice? What is the training time compared to SSR?

6. Scalability study. Why is ResNet-50 used instead of stronger backbones (e.g., ResNet-101, ViT)? Is the improvement backbone-agnostic? Is it possible for such framework to scale up with larger model size?

7. Why is closed-loop success rate on Bench2Drive still low (26.54%)? What are the main failure modes in Bench2Drive?

---

### Official Review · Reviewer_mwLw · 2025-11-01

**Soundness:** 3
**Presentation:** 3
**Contribution:** 2
**Rating:** 4
**Confidence:** 4

**Summary:**

This paper proposes Echo Planning (EchoP), a self-supervised end-to-end autonomous driving framework that enforces temporal consistency between predicted trajectories and scene dynamics through a future–past cycle consistency. The model predicts the future BEV representation and then the current BEV representation, penalizing inconsistencies via a cycle-consistency loss. Experiments on nuScenes and Bench2Drive demonstrate improvements over strong baselines such as UniAD and SSR with gains in open and closed-loop driving metrics.

**Strengths:**

- Introduces a cycle-consistency self-supervision scheme to enhances temporal coherence with bidirectional consistency in the planning process.
- Demonstrates improvements across open- and closed-loop evaluations, supported by ablations on loss weighting and token size.
- The formulation is conceptually intuitive and integrates easily into existing sparse token planning frameworks such as SSR.

**Weaknesses:**

- The paper omits discussion of recent, related works on temporal consistency in end-to-end planning, notably BridgeAD (CVPR 2025) and MomAD (CVPR 2025). While these methods differ in formulation (historical aggregation, momentum modeling) they share similar motivations.
- The conceptual advance over SSR is modest. It is unclear whether explicit cycle consistency is essential, as no alternative reconstruction formulations were tested. For instance, directly predicting the previous BEV (t−1) or a further future BEV (t+2).
- Open-loop performance gains are minor and do not clearly demonstrate stronger generalization beyond the SSR baseline.
- Computational cost and latency metrics are not reported.

**Questions:**

- How does EchoP performance compare to BridgeAD and MomAD, which also aim to enforce temporal consistency in dense planning?
- How does EchoP perform under the temporal prediction consistency (TPC) metric introduced in MomAD?
- Does EchoP introduce any additional inference latency relative to SSR?
- Could similar consistency be achieved through directly predicting the previous BEV frame  (t-1) or a further future BEV (t+2), without the explicit cycle loss?

---

### Note · Authors · 2025-11-28

I have read and agree with the venue's withdrawal policy on behalf of myself and my co-authors.